# Dynomics: A Novel and Promising Approach for Improved Breast Cancer Prognosis Prediction

**DOI:** 10.3390/jpm13061004

**Published:** 2023-06-15

**Authors:** Marianna Inglese, Matteo Ferrante, Tommaso Boccato, Allegra Conti, Chiara A. Pistolese, Oreste C. Buonomo, Rolando M. D’Angelillo, Nicola Toschi

**Affiliations:** 1Department of Biomedicine and Prevention, University of Rome Tor Vergata, 00133 Rome, Italy; matteo.ferrante@uniroma2.it (M.F.); tommaso.boccato@uniroma2.it (T.B.); allegra.conti@uniroma2.it (A.C.); chiarapistolese@gmail.com (C.A.P.); rolandomaria.dangelillo@ptvonline.it (R.M.D.); toschi@med.uniroma2.it (N.T.); 2Department of Surgery and Cancer, Imperial College London, London W12 0HS, UK; 3Diagnostic Imaging, Policlinico Tor Vergata, 00133 Rome, Italy; 4U.O.S.D. Breast Unit, Department of Surgical Science, Policlinico Tor Vergata, 00133 Rome, Italy; oreste.buonomo@uniroma2.it; 5Radiation Oncology, Policlinico Tor Vergata, 00133 Rome, Italy; 6Department of Radiology, Athinoula A. Martinos Center for Biomedical Imaging, Boston, MA 02129, USA

**Keywords:** breast cancer, prognosis, machine learning, radiomics, classification

## Abstract

Traditional imaging techniques for breast cancer (BC) diagnosis and prediction, such as X-rays and magnetic resonance imaging (MRI), demonstrate varying sensitivity and specificity due to clinical and technological factors. Consequently, positron emission tomography (PET), capable of detecting abnormal metabolic activity, has emerged as a more effective tool, providing critical quantitative and qualitative tumor-related metabolic information. This study leverages a public clinical dataset of dynamic ^18^F-Fluorothymidine (FLT) PET scans from BC patients, extending conventional static radiomics methods to the time domain—termed as ‘Dynomics’. Radiomic features were extracted from both static and dynamic PET images on lesion and reference tissue masks. The extracted features were used to train an XGBoost model for classifying tumor versus reference tissue and complete versus partial responders to neoadjuvant chemotherapy. The results underscored the superiority of dynamic and static radiomics over standard PET imaging, achieving accuracy of 94% in tumor tissue classification. Notably, in predicting BC prognosis, dynomics delivered the highest performance, achieving accuracy of 86%, thereby outperforming both static radiomics and standard PET data. This study illustrates the enhanced clinical utility of dynomics in yielding more precise and reliable information for BC diagnosis and prognosis, paving the way for improved treatment strategies.

## 1. Introduction

Breast cancer (BC) ranks as the second most prevalent cancer worldwide and the most common disease affecting women. Despite advances in screening programs and early-stage diagnosis contributing to improved survival rates, BC remains the sixth leading cause of death among females [1]. Key challenges in BC patient care involve diagnosis, tumor biology characterization, staging, therapeutic response, and prognosis prediction. Common diagnostic imaging techniques for breast cancer screening include breast MRI, ultrasound, and mammography. However, the sensitivity and specificity of these modalities in breast cancer categorization and prediction are limited due to various clinical and technological factors [2]. X-ray mammography, considered the gold standard for early breast cancer detection, suffers from a high rate of false positives [3]. Similarly, MRI faces limitations stemming from factors such as magnetic field strength, gradient strength, and coil performance [2]. While computed tomography (CT) and MRI detect anatomic changes for cancer diagnosis, staging, and follow-up, positron emission tomography (PET) captures abnormal metabolic activity, offering crucial qualitative and quantitative tumor-related metabolic information [4]. The most frequently used PET imaging tracer, ^18^F-Fluorodeoxyglucose (FDG), measures glucose metabolism and correlates with tumor proliferation [5]. The clinical validity of serial ^18^F-FDG-PET/CT to monitor therapy response to neoadjuvant treatment was analyzed in two meta-analyses which show a pooled sensitivity of 82–86% and a specificity of 72–79%, using histopathology as a reference standard for pathological (non-)response [6]. However, FDG is also linked to processes such as inflammation, cellular repair, and apoptosis, resulting in a high rate of false positives [7]. In fact, attempts to integrate ^18^F-FDG-PET/CT in the Response Evaluation Criteria in Solid Tumors (RECIST) criteria have not been successful so far, and ^18^F-FDG-PET/CT is not routinely used for response evaluation in BC due to the absence of sufficient clinical validation data [6,8]. These limitations have spurred the investigation of alternative PET imaging agents. ^18^F- fluoroestradiol (FES)-PET/CT enables the visualization of estrogen receptor (ER) expression, with ^18^F-FES behaving very similar to estradiol [9]. A meta-analysis of nine studies (all prospective, except one) involving 238 patients reported a pooled sensitivity of 82% and a specificity of 95% to detect ER+ tumor lesions by quantitative assessment of ^18^F-FES uptake. A similar sensitivity and specificity was found in direct comparison of ^18^F-FES uptake and ER expression on biopsy (in five studies including 158 BC patients) [6,10]. However, consistent data to support the clinical validity and utility of ^18^F-FES are still lacking [6]. The ^89^Zr-labeled antibody trastuzumab binds to the human epidermal growth factor (HER2) receptor and has a relatively long half-life [6]. At present, in a prospective study including 34 HER2+ and 16 HER2− BC patients, a standardized uptake value (SUV)max cut-off of 3.2 showed a sensitivity of 76% and a specificity of 62% to distinguish HER2+ from HER2− lesions [11]. ^18^F-Fluorothymidine (FLT), a labeled thymidine analogue, has gained interest for its potential in visualizing and quantifying cell proliferation, demonstrating a correlation with Ki-67 in breast, lung, and brain cancer [12]. Sanghera et al.’s comprehensive review highlights the high reproducibility of ^18^F-FLT-PET scans and the ability to differentiate between complete response (CR), partial response (PR), and stable disease (SD) using SUV changes at 90 min and Ki [13]. Additionally, ^18^F-FLT has been proposed as a proliferative indicator capable of quantifying tumor cell viability during or upon the onset of treatment [14]. Typically, these assessments use SUV data derived from static ^18^F-FLT PET images, offering only a general overview of uptake [15]. Factors such as metabolism, hypoxia, necrosis, and cell proliferation cause tracer absorption within a tumor mass to vary significantly, with this heterogeneity seemingly correlating with prognosis, treatment response, and tumor aggressiveness [16,17]. In vivo spatiotemporal tracer concentration maps, obtained from dynamic PET acquisitions, reflect tissue-specific biochemical properties and encompass information about target interaction and washout effects [18]. Dynamic PET images are predominantly used for research purposes, with their clinical application hindered by lengthy acquisition times (potentially uncomfortable for patients) and the need for compartmental modeling approaches to quantify tracer uptake. These modeling approaches improve tumor characterization and treatment response monitoring but require not only a deep understanding of the mathematical models but also invasive measurement of the arterial input function. Radiomics, an approach that quantifies lesion heterogeneity using medical imaging, has emerged as a promising research area in breast cancer [19]. Beyond traditional quantitative variables employed in radiology and nuclear medicine, such as dimensions, uptake, or volume factors, radiomics extracts an extensive array of quantitative features from medical images. Machine learning techniques are typically employed to manage the vast amount of data generated by radiomics. Most studies investigate the correlation between the texture parameters and immunohistochemical subtypes of breast cancer. Interestingly, one study reported no discriminative power for PET-derived texture metrics [20], while another found that radiomic performance improved when combined with clinicopathological features (AUC 0.80 vs. 0.73, *p* = 0.007) [20,21]. In this study, we aim to integrate radiomics with dynamic ^18^F-FLT PET images to assess the response to neoadjuvant chemotherapy (NAC) in BC patients, employing an approach we term “dynomics”, i.e., dynamic radiomics. While radiomics techniques are commonly applied to static PET data, we propose that extending these techniques to the time domain can potentially extract more clinically valuable information from the dynamic PET signal. By applying dynomics to PET data, we aim to overcome the limitations of standard compartmental modeling approaches while still capitalizing on the benefits of dynamic ^18^F-FLT PET acquisition.

## 2. Materials and Methods

### 2.1. Dataset

We utilized a publicly available clinical ^18^F-FLT PET dataset consisting of 44 breast cancer patients (for whom a dynamic PET scan was available at baseline and who included 19 partial (PR) and 12 complete responders (CR) to NAC), which is part of the “ACRIN-FLT-Breast (ACRIN 6688)” collection in the Cancer Imaging Archive (TCIA) (Table 1) [22,23,24]. No patient showed lymph node involvement. The inclusion criteria included: pathologically confirmed breast cancer, determined to be a candidate for primary systemic (neoadjuvant) therapy and for surgical resection of residual primary tumor following completion of neoadjuvant therapy; locally advanced breast cancer, not stage IV, and with a tumor size ≥ 2 cm (as measured on imaging or estimated by physical examination); no obvious contraindications for primary chemotherapy; the presence of a residual tumor planned to be removed surgically following completion of neoadjuvant therapy; the ability to lie still for 1.5 h for PET scanning; age 18 years or older; an Eastern Cooperative Oncology Group (ECOG) Performance Status ≤ 2 (Karnofsky ≥ 60%); a normal organ and marrow function as defined below during the first visit (leukocytes ≥ 3000/μL; absolute neutrophil count ≥ 1500/μL; platelets ≥ 100,000/μL; total bilirubin within normal institutional limits; AST(SGOT)/ALT(SGPT) ≤ 2.5 times the institutional upper limit of normal; and creatinine within normal institutional limits or creatinine clearance ≥ 30 mL/min/1.73 m^2^ for patients with creatinine levels above institutional normal); if female, postmenopausal for a minimum of one year, or surgically sterile, or not pregnant; and the ability to understand and willing to sign a written informed consent document and a Health Insurance Portability and Accountability Act (HIPAA) authorization in accordance with institutional guidelines. The exclusion criteria included: previous treatment (chemotherapy, radiation, or surgery) on the involved breast, including hormone therapy; an uncontrolled intercurrent illness including, but not limited to, ongoing or active infection, symptomatic congestive heart failure, unstable angina pectoris, cardiac arrhythmia, or psychiatric illness/social situations that would limit compliance with the study requirements; medical instability; a condition requiring anesthesia for PET scanning and/or unable to lie still for 1.5 h; a history of allergic reactions attributed to compounds of similar chemical or biologic composition to ^18^F-FLT; age under 18; pregnancy or nursing as the effects of ^18^F-FLT in pregnancy are not known; previous malignancy, other than basal cell or squamous cell carcinoma of the skin or in situ carcinoma of the cervix, from which the patient has been disease free for less than 5 years; and currently on hormone therapy as the primary systemic neoadjuvant therapy [23]. Dynamic PET images were acquired following a bolus injection of 167 MBq (mean; range, 110–204 MBq) using a General Electric (GE)/Philips Medical System PET/CT system. Dynamic scans (matrix dimension: 128 × 128 × 35; voxel dimensions in the x, y, and z axis: 3.9, 3.9, and 4.2) consisted of 45 timeframes (16 × 5, 7 × 10, 5 × 30, 5 × 60, 5 × 180, and 6 × 300 s) and a 60-min acquisition duration (mean, 70 min; range, 50–101 min). PET images were reconstructed using the CT data for attenuation correction with an ordered-subset expectation maximization iterative reconstruction algorithm (2 iterations and 28 subsets). All patients were scanned on calibrated and ACRIN-accredited PET/CT scanners, which underwent image quality review and SUV testing using a uniform phantom [23]. Only baseline scans were used in this study.

### 2.2. PET Data Pre-Processing

For each patient, an experienced radiologist manually contoured consecutive regions of interest around the tumor on the static PET image (obtained as the average of the last five timeframes of the dynamic PET data). The ^18^F-FLT radioactivity concentrations within the volumes of interest were normalized to the injected radioactivity and patient body weight to obtain SUV values [25]. Additionally, a region of interest was obtained from the centroid of the healthy contralateral breast, where the same mask obtained from lesion segmentation was flipped and used for delineating a reference healthy region, as described in [26].

### 2.3. Radiomic Feature Extraction

Radiomic features were extracted within the lesion and reference tissue VOIs using Python software and the Pyradiomics module. The features included first order, shape (2D), shape (3D), gray level cooccurrence matrix (GLCM), gray level size zone matrix (GLSZM), gray level run length matrix (GLRLM), neighboring gray tone difference matrix (NGTDM), and gray level dependence matrix (GLDM). A total of 107 features were extracted from the static PET image (static radiomics) and within each frame of the dynamic ^18^F-FLT PET acquisition (dynamic radiomics). In the latter case, the median and median absolute deviation (MAD) across time were computed. The full list of extracted features is included in Appendix A.

### 2.4. Machine Learning Models

We employed an XGBoost model in a five-fold stratified and nested cross-validation manner. This included hyperparameter optimization (maximum depth of a tree, minimum sum of instance weight needed in a child, and subsample ratio of columns for each tree) within the inner loop and performance quantification (outer loop) for two classification tasks: (1) tumor vs. reference tissue and (2) CR vs. PR. Within each inner fold, principal component analysis (PCA) was employed to reduce the dimensionality of the dataset after a standardization step (all transformations were computed on the training set and applied onto the test set). The extraction through PCA was set to account for 90% of the total variance in each dataset.

The input for the XGBoost model in each classification task was an (n × m) matrix of radiomic features, with n being the sample size (n = 88 for tumor vs. reference tissue classification, and n = 31 for CR vs. PR classification) and m being the number of features (principal components resulting from PCA) when using static radiomics, and median, MAD, and (median + MAD) dynamic radiomics feature values. For the latter, the median and MAD feature values were concatenated in an (n × 2 m) matrix. The performance of the XGBoost model was also compared to the deep learning models that we tested and validated in [26], which used static and dynamic PET images for the same classification tasks. For static PET images, we employed the CONV3D model based on three-dimensional convolutional filters, taking an (n × im_x × im_y × im_z) matrix as input, with im_x, im_y, and im_z being the dimensions of the static PET image (reduced to a box with dimensions im_x = 30, im_y = 30, and im_z = 10, as detailed in [26]). For dynamic PET images, we combined three-dimensional convolutional layers with long short-term memory (LSTM) filters in the CONV3D + LSTM model, which took an (n × im_x × im_y × im_z × t) matrix as input, with t representing the number of frames in the dynamic PET image (t = 45). In addition, the performance of the XGBoost model employing “summary” dynamic radiomics features (median, MAD feature values) was compared to a deep learning model based on mono-dimensional convolutional layers (the CONV1D model) to extract meaningful information from the time evolution of each feature. The input to the CONV1D model was an ((n × m) × t) matrix, where the time evolution of each feature was labeled individually. Model performance is reported in terms of the area under the receiver operating characteristic (ROC) curve (AUC), accuracy, precision, and recall [27]. To examine the unique contributions of each feature to the final prediction, we calculated the SHapley Additive exPlanations (SHAP) values for each model [28]. All experiments were conducted using Python version 3.8, the Keras deep learning library, with TensorFlow as the backend. A Linux machine and two Nvidia Pascal TITAN V GPUs with 12 GB RAM each were employed. The workflow for image acquisition, segmentation, feature extraction, and classification model assessment is shown in Figure 1.

## 3. Results

Figure 2 displays the correlation matrices of both the static radiomics features and the median and MAD dynamic feature values. Due to the significant intercorrelation among the variables, as mentioned above, we leveraged principal component analysis to reduce the dimensionality of static radiomics to eight and five principal components in the tumor vs. reference tissue and CR vs. PR classification tasks, respectively. Similarly, for the median, MAD, and median + MAD feature values derived from dynomics, the dimensionality was reduced to 7, 9, and 10 (tumor vs. reference tissue) and 5, 5, and 6 (CR vs. PR) principal components. In all PCA procedures, extraction was set to account for 90% of the total variance in the respective dataset.

### 3.1. Tumour vs. Reference Tissue Classification

Table 2 provides a summary of the results obtained when classifying tumor tissues using static and dynamic radiomic features. The performance of the XGBoost model, which utilized static and dynamic radiomics, was compared to the CONV3D and CONV3D + LSTM models, which employed static and dynamic PET images for classification. Table 2 also presents the results obtained when utilizing median, MAD, and median + MAD computed across time (input to the XGBboost model) or the temporal evolution of each extracted feature (input to the CONV1D model) for classification under the dynamic radiomics framework. Our results indicate that radiomics, both static and dynamic, outperforms standard PET image use, achieving an impressive 94% accuracy (AUC: 0.94), in contrast to 61% accuracy (AUC: 0.59) and 75% accuracy (AUC: 0.81) obtained with 3D and 4D PET data, respectively (Table 2). Additionally, the performance of the XGBoost model (which utilized median (94% accuracy, 0.94 AUC), MAD (89% accuracy, 0.89 AUC) and median + MAD (94% accuracy, 0.94 AUC) dynamic radiomics feature values) was superior to the CONV1D model, which achieved 49% accuracy and an AUC: 0.49 (Table 2).

Figure 3A–C presents the feature importance as determined by SHAP for static radiomics, which demonstrated optimal performance related to tumor vs. reference tissue classification. As indicated in Figure 3A,B, the first principal component (PC1) derived from PCA consistently holds the greatest explanatory power. Figure 3C’s table enumerates the top five original features that significantly influence the primary contributing component (PC1), along with the eigenvalues. These features all exhibit positive eigenvalues, implying a direct proportionality with the component’s score. They fall within first-order statistics and include the root mean square, the 90th percentile, and the mean uptake of ^18^F-FLT. The final two crucial features for classification are GLCM joint average and sum average, which elucidate the lesion’s texture. GLCM features encapsulate the second-order statistical data of gray levels between adjacent pixels in an image [29]. Joint and mean averages denote the average gray level sum distribution of the image. Collectively, these top five features suggest that the average proliferative activity of the tumor, as signified by the uptake of ^18^F-FLT and quantifiable by first and second order statistics, sufficiently distinguishes the varying thymidine activity between the tumor and healthy tissue.

### 3.2. Complete vs. Partial Responders Classification

Table 3 consolidates the results obtained when classifying treatment response using static and dynamic radiomic features. The performance of the XGBoost model, which utilized static and dynamic radiomics, was once more compared to the CONV3D and CONV3D + LSTM models. These models respectively used static and dynamic PET images for classification, akin to the previous set of analyses. Table 3 also elucidates the results obtained when using the median, MAD, and median + MAD computed across time for dynamic radiomics (input to the XGBboost model), or the temporal evolution of each extracted feature (input to the CONV1D model), for PR vs. CR classification. Our results show that dynamic radiomics achieved the highest performance, attaining 86% accuracy (AUC: 0.83). This approach significantly outperformed both static radiomics (71% accuracy, AUC: 0.67) and standard PET data (static: 59% accuracy, AUC: 0.51; dynamic: 60% accuracy, AUC: 0.59). Remarkably, the performance of the XGBoost model, which utilized median (71% accuracy, AUC: 0.67), MAD (57% accuracy, AUC: 0.58) and median + MAD (86% accuracy, AUC: 0.83) dynamic radiomics feature values, surpassed the CONV1D model, which achieved 52% accuracy and an AUC: 0.50.

Figure 3D–F delineates feature importance as computed by SHAP for median + MAD dynomics, the best-performing method related to partial vs. complete responders’ classification. As indicated in Figure 3D,E, the sixth principal component (PC6) derived from PCA consistently holds the greatest explanatory power. The table in Figure 3F identifies the top five original features that significantly influence the primary contributing component (PC6), accompanied by the eigenvalues. Notably, the Median GLSZM Size Zone Non-Uniformity and Low Gray Level Zone Emphasis features, along with the Median NGTDM Contrast, exhibit positive eigenvalues, whereas the Median GLCM Idn and the MAD GLDM Dependence Non-Uniformity Normalized features have negative eigenvalues. A Gray Level Size Zone measures gray level zones in an image, defined as the number of connected voxels with identical gray level intensity (specifically, the uptake of ^18^F-FLT in our case). High values of Size Zone Non-Uniformity and Low Gray Level Zone Emphasis features suggest less homogeneity in size zone volumes and a larger proportion of lower gray-level values and size zones in the image, respectively. GLCM Inverse Difference Normalized (IDN) offers another measure of an image’s local homogeneity by normalizing the difference between neighboring intensity values via division over the total count of discrete intensity values. These features, in conjunction with NGTDM Contrast, which relies on the overall gray level dynamic range of the image, suggest that the heterogeneous distribution of ^18^F-FLT uptake, mirroring the activity of thymidine within the lesion, serves as the key differentiator between partial and complete responders.

## 4. Discussion

In the realm of locally advanced breast cancer treatment, the established therapeutic strategy involves neoadjuvant chemotherapy followed by surgery, with an objective response rate hovering around 70% and a complete pathological response rate of nearly 30% [14,30]. Typically, the response evaluation rests on the histopathological examination of the surgical specimen. However, no universal consensus exists regarding the premier imaging method for early response assessment. Generally, morphological imaging techniques are employed, interpreted according to the Response Evaluation Criteria in Solid Tumors criteria (RECIST v1.1) [31]. Functional imaging methods are also utilized to evaluate the early response to therapy. The most widely used method to monitor therapeutic response in breast cancer is ^18^F-FDG, but it lacks high tumor-specificity as it also accumulates in activated macrophages and other inflammatory cells [14]. Thus, ^18^F-FLT has been suggested as a proliferation marker, enabling the quantification of the tumor’s proliferative activity and improving the assessment of tumor cell viability during or upon the commencement of treatment [12,15]. Typically, these evaluations are conducted using semi-quantitative SUV statistics, particularly SUVmax [15]. However, these SUV indices provide only a broad overview of uptake and fail to identify the presence of an uneven uptake distribution [14,26]. It is important to note that tracer absorption within a tumor mass is known to exhibit significant variability due to numerous factors, such as metabolism, hypoxia, necrosis, and cell proliferation, and this heterogeneity appears to be associated with prognosis, treatment response, and tumor aggressiveness [17]. These limitations warrant a more sophisticated approach, integrating dynamic PET data with radiomics.

Dynamic PET acquisition, which commences at the point of radiotracer injection and includes continuous acquisition of a PET bed position for several minutes to an hour or more, supplements the limited information gleaned from static PET acquisition [32,33,34,35]. Dynamic PET provides in vivo mapping of the spatiotemporal tracer concentration, accounting for the drug’s contact and washout effects with the target. The current role of PET radiomics in breast cancer has been recently summarized by Urso and colleagues, who identified 81 studies, of which 43 (81.1%) were retrospective and 10 (18.9%) were prospective [36]. In these studies, radiomic data were extracted exclusively from static PET images [36].

Our study introduces a novel concept of extending radiomic techniques to the time domain to extract more clinically pertinent information from the dynamic PET signal. Within our cohort of primary breast cancer patients undergoing NAC, we extracted radiomic features from both standard static PET images and each frame of the dynamic PET image. We were able to perform two classification tasks, achieving accuracy of 94% (0.94 AUC) for tumor tissue classification and 86% accuracy (0.83 AUC) for partial vs. complete response to treatment, outperforming both static radiomics and standard PET image use.

Figure 3 delineates the feature importance calculated via SHAP for the best-performing model for each task. The efficacy of static radiomics (or simply radiomics) in tumor tissue classification was significantly influenced by the first principal component (PC1), computed through PCA. Particularly, first and second order statistical features, reflective of the increased proliferative activity of a breast lesion compared to healthy reference tissue, were key. For the task of classifying complete vs. partial responders, the sixth component accounted for the highest explained variance. Specifically, features describing the level of heterogeneity within the lesion emerged as the most critical for this classification task. This finding implies that the presence of lesion heterogeneity, due to subregions characterized by distinct proliferative activity, is a hallmark of tumors that partially responded to NAC compared to complete responders. This result corroborates numerous published studies asserting that tumor heterogeneity is the primary challenge in cancer therapy. Further comprehension of both non-genetic and genetic aspects of tumor heterogeneity may provide a pathway to overcome therapeutic resistance and enhance cancer treatment [37].

Despite these promising results, one limitation of our study is the small sample size. To address this, we reduced the data dimensionality using PCA. Unfortunately, we could not perform any pharmacokinetic analysis or provide any comparison with kinetic parameters as the database we used did not include information about the percentage of the metabolite FLT-glucuronide present in the blood after the injection of the ^18^F-FLT tracer [38,39]. Hence, the parent plasma (metabolite-corrected) input function necessary for pharmacokinetic fitting was not available. An additional limitation of our study is related to the extraction of radiomic features from images acquired using different scanners. In fact, radiomic features are susceptible to variation across scanners, acquisition protocols, and reconstruction settings [40]. In addition, in the second prediction tasks, the slight class imbalance may have influenced our results, and due to the uniqueness of the dataset, we were not able to perform external validation on an equivalent set of data. Additional challenges which may have impacted the quality of the breast lesion segmentation could be image noise and artifacts as well as the heterogeneity of breast tumors themselves, also given that we did not have access to an additional specialist to verify user-dependent segmentation variability. In addition, due to the heterogeneity of techniques and data types/shapes, we chose not to use any data augmentation techniques.

Future studies with larger cohorts, multimodal imaging [41], and prospective designs are warranted to validate the results observed in this study. Moreover, incorporating other imaging modalities such as MRI [41,42] and CT, along with PET [43], may further improve the performance of radiomics in assessing tumor response to therapy. In addition, machine learning algorithms and deep learning models can be employed to refine feature extraction and data analysis, potentially leading to even better prediction accuracy [44,45]. Still, the concept of applying radiomics to the time domain has the potential to revolutionize the way we evaluate treatment response in breast cancer patients. By extracting meaningful features from both static and dynamic PET images, we can more accurately predict patient outcomes and tailor therapeutic interventions to individual needs. Furthermore, we would possibly investigate the potential role of preoperative ^18^F-FLT PET/CT dynomics in predicting hormone receptor (HR) positivity, as compared to the standard approach with ^18^F-FDG which showed unfavorable results [46].

In conclusion, this study represents a pioneering effort in the application of dynamic radiomics to ^18^F-FLT-PET data for the assessment of treatment response in breast cancer patients. Our findings demonstrate the superiority of this approach compared to conventional static radiomics, suggesting that dynamic radiomics holds great promise as a valuable tool in the management of breast cancer. By providing more accurate and detailed information regarding tumor response to therapy, dynomics can potentially lead to improved patient outcomes and the optimization of individualized treatment strategies.

## Figures and Tables

**Figure 1 jpm-13-01004-f001:**
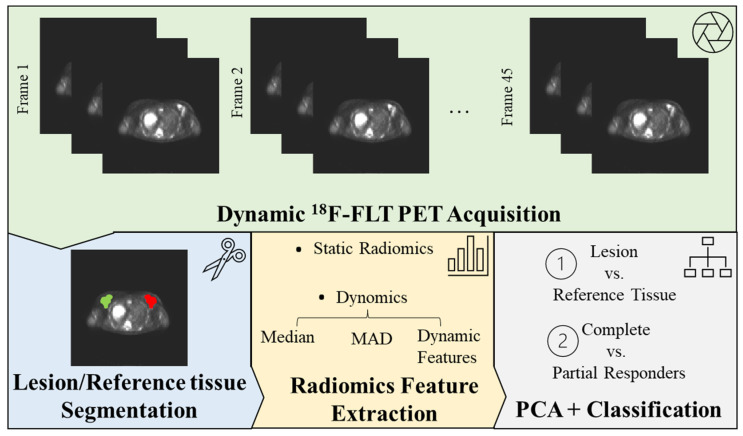
Radiomics Workflow. Beginning with the acquisition of medical images, volumes of interest (VOIs) are manually segmented on both the lesions and a healthy reference tissue. Radiomic features are subsequently extracted from these VOIs and superimposed onto the static ^18^F-FLT PET image (a 3D image derived from averaging the final five time-frames of the dynamic acquisition), thus establishing the basis for static radiomics. For our novel dynomics approach, these features are extrapolated from each frame of the dynamic ^18^F-FLT PET acquisition. In this process, summary values—including median and median absolute deviation (MAD)—are evaluated for each feature and analyzed in conjunction with their temporal evolution (dynamic features). The features encapsulate information about the tumor’s shape, first-order statistical features (derived from the image intensity histogram), and second-order statistical features (texture features). To optimize the data for interpretation, radiomics features undergo redundancy correction via principal component analysis (PCA), enabling the analysis of only non-redundant, meaningful features. These streamlined features are then processed through a machine learning model (XGBoost), generating a clinically interpretable outcome (lesion vs. reference tissue and complete vs. partial responders’ classification).

**Figure 2 jpm-13-01004-f002:**
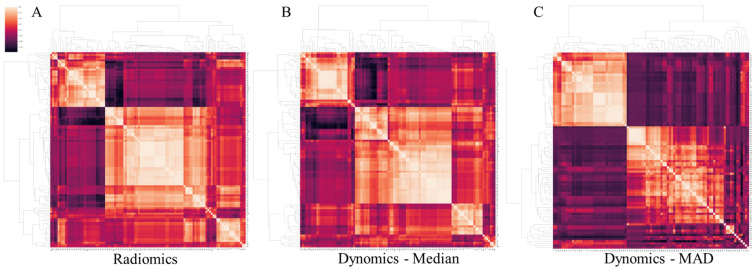
Feature correlation matrices, demonstrating high redundancy across the original static (**A**) and median (**B**) and mean absolute deviance (MAD) (**C**) of the dynamic radiomic features.

**Figure 3 jpm-13-01004-f003:**
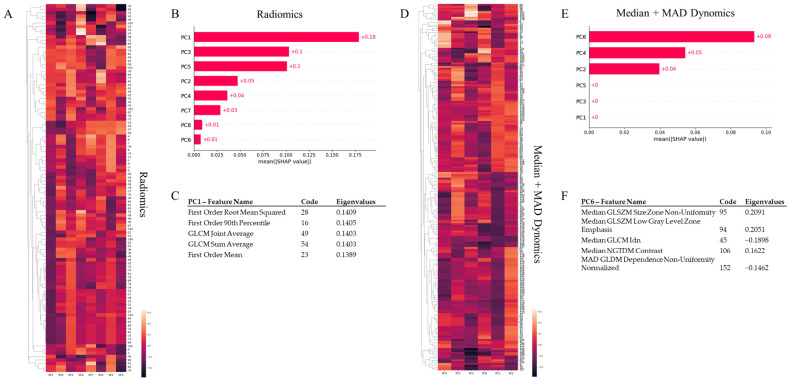
PCA and SHAP analysis for radiomics and median + MAD dynomics. Given the substantial intercorrelation among variables, principal component analysis (PCA) was employed to condense the dimensionality of static radiomics to eight principal components for tumor vs. reference tissue comparison and median + MAD dynomics to six principal components for CR vs. PR classification. For all PCA procedures, extraction was configured to account for 90% of the total variance in the respective dataset. SHapley Additive exPlanation (SHAP) was utilized to quantify the contribution of each principal component, derived through PCA, to the classification model. (**A**,**D**) The heatmaps illustrate, for each principal component used to classify tumor vs. reference tissue using radiomics (**A**) and PR vs. CR employing median + MAD dynomics (**D**), the clustered eigenvalues evaluated for each feature. (**B**,**E**) These are variance importance plots detailing the most significant principal components, listed in descending order, contributing to the classification of tumor tissues using radiomics and PR using median + MAD dynomics. PC1 and PC6 were the most substantial contributors to each classification task, respectively. (**C**,**F**) These depict the top five features that load the highest on the component contributing the most to the classification task (radiomics—PC1 for tumor vs. reference tissue classification, and median + MAD dynomics—PC6 for CR vs. PR classification), along with the total explained variance.

**Table 1 jpm-13-01004-t001:** Patients’ demographics.

Demographic	Variable	Patients (n = 44)
Mean age ± SD (y)		52.5 ± 10.6
Mean tumor size ± SD (cm)		4.6 ± 2.5
Menopausal status	Premenopausal	16 (36.4%)
	Postmenopausal	28 (63.6%)
Initial diagnosis	Invasive breast cancer NOS	1 (2.27%)
	Invasive ductal	39 (88.63%)
	Invasive lobular + mixed invasive and lobular	4 (9.09%)
Estrogen receptor status	Positive	24 (54.54%)
	Negative	20 (45.45%)
Progesterone receptor status	Positive	25 (56.81%)
	Negative	29 (65.91%)
HER2 status	Positive	12 (27.27%)
	Negative	22 (50.00%)
Receptor status	Triple negative	9 (20.45%)
	Other	35 (79.54%)
Grade at diagnosis	1	1 (2.27%)
	2	9 (20.45%)
	3	27 (61.36%)

Percentages not adding up to 100% are due to missing data; tumor size was determined using baseline imaging. HER2 = human epidermal growth factor receptor type 2. SD = standard deviation.

**Table 2 jpm-13-01004-t002:** Summary of model performances when discriminating tumors from the reference tissue using static and dynamic radiomic features and images.

			AUC	Accuracy	Precision		Recall	
PET Acquisition	Input Data	Model			Lesion	Reference	Lesion	Reference
**Static**	PET image	CONV3D	0.59 (±0.09)	0.61 (±0.99)	0.56 (±0.13)	0.61 (±0.21)	0.67 (±0.17)	0.48 (±0.15)
	Radiomics	XGBoost	0.94	0.94	1.00	0.90	0.89	1.00
**Dynamic**	PET image	CONV3D + LSTM	0.81 (±0.08)	0.75 (±0.09)	0.69 (±0.09)	0.91 (±0.09)	0.96 (±0.03)	0.55 (±0.17)
	Dynomics—Median	XGBoost	0.94	0.94	0.90	1.00	1.00	0.89
	Dynomics—MAD	XGBoost	0.89	0.89	1	0.82	0.78	1.00
	Dynomics—Median + MAD	XGBoost	0.94	0.94	1	0.90	0.89	1
	Dynomics	CONV1D	0.49 (±0.05)	0.49 (±0.05)	0.49 (±0.03)	0.48 (±0.02)	0.84 (±0.11)	0.15 (±0.10)

**Table 3 jpm-13-01004-t003:** Summary of model performance when discriminating complete from partial responders using static and dynamic radiomic features and images.

			AUC	Accuracy	Precision		Recall	
PET Acquisition	Input Data	Model			CR	PR	CR	PR
**Static**	PET image	CONV3D	0.50 (±0.00)	0.59 (±0.18)	0.30 (±0.37)	0.20 (±0.26)	0.40 (±0.48)	0.60 (±0.48)
	Radiomics	XGBoost	0.67	0.71	1.00	0.67	0.33	1.00
**Dynamic**	PET image	CONV3D + LSTM	0.50 (±0.00)	0.59 (±0.18)	0.30 (±0.37)	0.29 (±0.27)	0.40 (±0.49)	0.60 (±0.49)
	Dynomics—Median	XGBoost	0.67	0.71	1.00	0.67	0.33	1.00
	Dynomics—MAD	XGBoost	0.58	0.57	0.50	0.67	0.67	0.50
	Dynomics—Median + MAD	XGBoost	0.83	0.86	1.00	0.80	0.67	1.00
	Dynomics	CONV1D	0.50 (±0.00)	0.52 (±0.11)	0.37 (±0.30)	0.15 (±0.19)	0.60 (±0.49)	0.40 (±0.49)

## Data Availability

A publicly available clinical ^18^F-FLT PET dataset consisting of 44 breast cancer patients (19 partial (PR) and 12 complete responders (CR) to NAC), which is part of the “ACRIN-FLT-Breast (ACRIN 6688)” collection in the Cancer Imaging Archive (TCIA) [22,23,24] was used. Link: https://wiki.cancerimagingarchive.net/pages/viewpage.action?pageId=30671268, accessed on 1 March 2022.

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
