# Peer review of "Dynomics: A Novel and Promising Approach for Improved Breast Cancer Prognosis Prediction"

_jpm, 2023, doi:10.3390/jpm13061004_

Round 1
Reviewer 1 Report
Congratulations to the authors, it is an interesting study that reflects the need of improvement with new tech to BC imaging with machine learning methods.
After reading the paper I had some questions to the authors. The answers could be adapted and incorporated in the text, mainly to complement the Method and in the Discussion as limitations of the study:
- What machine learning algorithm(s) were employed in this study?
- How was the training dataset for the machine learning models constructed?
- Did the authors perform any feature selection or feature engineering techniques before applying the machine learning models?
- Were there any preprocessing steps applied to the input data before training the machine learning models?
- What evaluation metrics were used to assess the performance of the machine learning models?
- Were any cross-validation or resampling techniques utilized to estimate the model's performance?
- Were there any considerations or adjustments made to handle class imbalance, if present in the dataset?
- What was the rationale behind selecting the specific radiomic features for model input?
- Did the authors provide any insights into the computational resources required for training and testing the machine learning models?
- Were any efforts made to validate the generalizability of the machine learning models, such as external validation on an independent dataset?
- Were any specific inclusion or exclusion criteria applied when selecting patients for this study?
- Did the study account for the variability in imaging techniques or protocols across different imaging centers?
- How were the radiomic features extracted from the static and dynamic PET images? Was any software or algorithm used for this purpose?
- Were there any challenges or limitations encountered in accurately delineating the regions of interest (ROIs) for radiomic analysis in breast tumors?
- Were there any specific considerations or adjustments made for patients with breast implants or other anatomical variations?
- Did the study assess the interobserver or intraobserver variability in radiomic feature extraction to ensure reliability and reproducibility?
- Was there any analysis or discussion regarding the potential impact of hormonal receptor status (estrogen receptor, progesterone receptor) on the radiomic features and machine learning models?
- Were there any correlations or comparisons made between the radiomic features and clinical-pathological factors, such as tumor size, lymph node involvement, or histological grade?
- Did the study investigate the potential influence of neoadjuvant chemotherapy regimens or treatment response on the radiomic features and machine learning models? Turn more clear in the text.
- Were there any discussions or insights provided regarding the potential clinical implications or future directions of incorporating dynamic radiomics in the routine assessment of breast cancer treatment response?
The overall quality of the English language in this article is good. The text demonstrates a proficient use of academic English, with clear and concise sentences and appropriate vocabulary for the subject matter. The author effectively communicates complex ideas and concepts related to breast cancer and machine learning.
The article is well-structured, with distinct sections such as Introduction, Method, Results, Discussion, and Conclusion, which enhance readability and organization. The language used in each section appropriately matches the purpose and content expected in scientific writing.
Throughout the article, the author consistently maintains a formal tone and adheres to the conventions of academic writing. The grammar and syntax are generally accurate, with only a few minor errors or inconsistencies. The use of punctuation, such as commas and parentheses, is appropriate, aiding in the clarity and coherence of the text.
The article demonstrates a wide range of medical and technical vocabulary, reflecting the author's expertise in the field. The terminology used is precise and specific, contributing to the clarity and accuracy of the scientific content.
In terms of style, the author effectively uses transitional phrases and cohesive devices to connect ideas and facilitate the flow of information. The logical progression of the arguments and the use of evidence from previous studies contribute to the overall cohesiveness of the article.
Overall, the article exhibits a high level of proficiency in the English language, effectively conveying scientific information related to breast cancer and machine learning.
Author Response
Please see the attachment.
Revision n.1, Inglese M et. Al, “Dynomics: a novel and promising approach for improved breast cancer prognosis prediction”
Before we begin the point-by-point reply to the reviewer's comments, we would sincerely like to thank the Editor and Editorial Board for the opportunity to resubmit a revised version of our paper, which has been revised in accordance with all reviewer's suggestions.
We would also like to thank the Reviewers for their very insightful comments and feedback on our manuscript. We greatly appreciate your valuable suggestions for enhancing the quality of our research. We have addressed each point as detailed below. We are confident that the revised version of the Manuscript is considerably improved with respect to the first submission and hope that the Reviewers will be available to re-read our work in this respect. All changes in the revised version of the manuscript are highlighted in tracked changes.
Reviewer 1
Congratulations to the authors, it is an interesting study that reflects the need of improvement with new tech to BC imaging with machine learning methods.
Reply: Thank you very much for appreciating our work.
After reading the paper I had some questions to the authors. The answers could be adapted and incorporated in the text, mainly to complement the Method and in the Discussion as limitations of the study:
What machine learning algorithm(s) were employed in this study?
Reply: This information is included in paragraph 2.4, “Machine Learning Models”, and has been rephrased for clarity in the revised version of the manuscript. Briefly, we employed an XGBoost model for two classification tasks: 1) tumor vs. reference tissue, and 2) complete vs partial responders.
How was the training dataset for the, “Machine Learning Models”, constructed?
Reply: As stated in paragraph 2.4 of the revised version of the manuscript “Machine Learning Models”, we employed an XGBoost model in a 5-fold nested cross-validation manner. Within each loop, four fifths of the sample were used for training and the rest for testing.
Did the authors perform any feature selection or feature engineering techniques before applying the machine learning models?
Reply: This information is included in paragraph 2.4, “Machine Learning Models”, and has been rephrased for clarity. A principal component analysis (PCA) was employed to reduce data dimensionality. PCA was applied only to training sets, and the transformation matrix was subsequently applied to test sets.
Were there any preprocessing steps applied to the input data before training the machine learning models?
Reply: Thank you for this interesting question. Data standardization was the only pre-processing step applied before training the models. In the revised version of the manuscript, this information has been no included in paragraph 2.4.
What evaluation metrics were used to assess the performance of the machine learning models?
Reply: This information is included in paragraph 2.4, “Machine Learning Models”, and has been rephrased for clarity in the revised version of the manuscript. Model performances are reported in terms of area under the receiver operating characteristic (ROC) curve (AUC), accuracy, precision, and recall.
Were any cross-validation or resampling techniques utilized to estimate the model's performance?
Reply: This information is included in paragraph 2.4, “Machine Learning Models”, and has been rephrased for clarity. The XGBoost model was employed in a 5-fold nested cross-validation manner.
Were there any considerations or adjustments made to handle class imbalance, if present in the dataset?
Reply: Thank you for this interesting question. In the first task, classes were perfectly balanced. In the second tasks, a slight imbalance was present. Although nested cross-validation was performed in a stratified fashion, class imbalance may still have impacted our results to some extent. The extent of this impact can be qualitatively judged by inspecting classification metrics like precision and recall. In the revised version of the manuscript, this has been discussed as a potential limitation.
What was the rationale behind selecting the specific radiomic features for model input?
Reply: We relied on previous literature and extracted all features provided in the popular package PyRadionmics (107 features) from the static PET image and within each frame of the dynamic PET image (as detailed in paragraph 2.3 Radiomic Feature Extraction). However, upon inspection of the covariance matrix we decided to reduce this data by projecting into an orthogonal space. We therefore applied a principal component analysis. The generated vectors were selected to explain the 90% of the total variance of the sample. In the revised version of the manuscript, this information has been included in paragraph 2.4.
Did the authors provide any insights into the computational resources required for training and testing the machine learning models?
Reply: This information is included in paragraph 2.4, “Machine Learning Models”, and has been rephrased for clarity in the revised version of the manuscript. All experiments were conducted using Python version 3.8, the Keras deep learning library, with TensorFlow as the backend. We employed a Linux machine with 256 GB RAM, and two Nvidia Pascal TITAN V GPU cards with 12 GB RAM each.
Were any efforts made to validate the generalizability of the, “Machine Learning Models”, such as external validation on an independent dataset?
Reply: Thank you for this insightful comment. We employed a public dataset (ACRIN 6688) which was the only data available to us. The potential usefulness of external validation / validation in a separate dataset has been discussed in n the revised version of the manuscript.
Were any specific inclusion or exclusion criteria applied when selecting patients for this study?
Reply: This study analyses data from the ACRIN-FLT-Breast (ACRIN 6688) “collection in The Cancer Imaging Archive (TCIA). Inclusion criteria included: a pathologically confirmed breast cancer, determined to be a candidate for primary systemic (neoadjuvant) therapy and for surgical resection of residual primary tumor following completion of neoadjuvant therapy; a locally advanced breast cancer, not stage IV, and with a tumor size ≥ 2cm (as measured on imaging or estimated by physical exam); no obvious contraindications for primary chemotherapy; the presence of a residual tumor planned to be removed surgically following completion of neoadjuvant therapy; the ability to lie still for 1.5 hours for PET scanning; age 18 years or older; an Eastern Cooperative Oncology Group (ECOG) Performance Status ≤ 2 (Karnofsky ≥ 60%); a normal organ and marrow function as defined below at first visit (leukocytes ≥ 3,000/μl; absolute neutrophil count ≥ 1,500/μl; platelets ≥ 100,000/μl; total bilirubin within normal institutional limits; AST(SGOT)/ALT(SGPT) ≤ 2.5 times the institutional upper limit of normal; creatinine within normal institutional limits or creatinine clearance ≥ 30 mL/min/1.73 m2 for patients with creatinine levels above institutional normal); if female, postmenopausal for a minimum of one year, OR surgically sterile, OR not pregnant; the ability to understand and willing to sign a written informed con-sent document and a Health Insurance Portability and Accountability Act (HIPAA) authorization in accordance with institutional guidelines. Exclusion criteria included: previous treatment (chemotherapy, radiation, or surgery) to involved breast; including hormone therapy; an uncontrolled intercurrent illness including, but not limited to, ongoing or active infection, symptomatic congestive heart failure, unstable angina pectoris, cardiac arrhythmia, or psychiatric illness/social situations that would limit compliance with study requirements; medical instability; a condition requiring anesthesia for PET scanning and/or unable to lie still for 1.5 hours; a history of allergic reactions attributed to compounds of similar chemical or biologic composition to 18F-FLT; age under 18; pregnancy or nursing as the effects of 18F-FLT in pregnancy are not known; previous malignancy, other than basal cell or squamous cell carcinoma of the skin or in situ carcinoma of the cervix, from which the patient has been disease free for less than 5 years; currently on hormone therapy as the primary systemic neoadjuvant therapy.
In the revised version of the Manuscript, this information has been included in Paragraph 2.1. We also clarified that patients were selected based on the availability of a dynamic 18F-FLT PET scan at baseline.
Did the study account for the variability in imaging techniques or protocols across different imaging centers?
Reply: As also suggested by Reviewer 2, we address the limitation of using radiomic features extracted from images acquired on different scanners in the Discussion section. However, as stated in paragraph 2.1, all patients were scanned on calibrated and ACRIN-accredited PET/CT scanners, which underwent image quality review and standardized uptake value testing using a uniform phantom. We also rephrased this part for the sake of clarity in the revised version of the Manuscript.
How were the radiomic features extracted from the static and dynamic PET images? Was any software or algorithm used for this purpose?
Reply: This information is included in paragraph 2.3, and has been rephrased for clarity in the revised version of the Manuscript. Radiomic features were extracted within the lesion and reference tissue VOIs using Python software and the Pyradiomics module. They were extracted from the static PET image and within each frame of the dynamic PET image.
Were there any challenges or limitations encountered in accurately delineating the regions of interest (ROIs) for radiomic analysis in breast tumors?
Reply: thank you for allowing us to expand on this matter. Common challenges, which we occasionally encountered, are a) Heterogeneity of Breast Tumors (Breast tumors can vary greatly in size, shape, and density. This heterogeneity can make it difficult to accurately delineate the ROI, particularly for small or irregularly shaped tumors) and Image Noise and Artifacts (This is particularly a problem for certain imaging modalities such as mammography, where artifacts like microcalcifications or breast implants can interfere with the tumor delineation). This information has been included in the discussion section of the revised version of the paper.
Were there any specific considerations or adjustments made for patients with breast implants or other anatomical variations?
Reply: There was no patient included in the study with breast implants or other anatomical variations (as expressed in paragraph 2.1 among the inclusion/exclusion criteria).
Did the study assess the interobserver or intraobserver variability in radiomic feature extraction to ensure reliability and reproducibility?
Reply: Thank you for making this important point. Only one expert radiologist participated in the delineation of the ROI’s, and the task was performed only once. We have included this aspect in the limitation section of our study.
Was there any analysis or discussion regarding the potential impact of hormonal receptor status (estrogen receptor, progesterone receptor) on the radiomic features and machine learning models?
Reply: We thank the reviewer for this comment and added a short paragraph in the Discussion section about the impact of hormonal receptor status on features and models. In the revised version of the Manuscript, We cite an interesting study by Araz and colleagues where preoperative 18F-FDG PET/CT radiomics were not able to predict the HR status of primary breast cancer. It would be interesting to investigate the role of 18F-FLT within this same task in a future study. This aspect has been included in the discussion section of the paper.
Were there any correlations or comparisons made between the radiomic features and clinical-pathological factors, such as tumor size, lymph node involvement, or histological grade?
Reply: In this pilot study, radiomic features were not compared to histological grade. Tumor size corresponds to one of the features extracted by the algorithm. No patients showed lymph node involvement. This information has been included in the revised version of the manuscript.
Did the study investigate the potential influence of neoadjuvant chemotherapy regimens or treatment response on the radiomic features and machine learning models? Turn more clear in the text.
Reply: One of the aims of this study was to test the ability of a novel radiomic approach for the prediction of prognosis. The information about the type of response to neoadjuvant chemotherapy was used to label each subject and to then test the performance of the machine learning model in the classification of partial vs complete responders. In the revised version of the text, this was clarified both in the Introduction and in the Discussion section.
Were there any discussions or insights provided regarding the potential clinical implications or future directions of incorporating dynamic radiomics in the routine assessment of breast cancer treatment response?
Reply: As discussed in paragraph 4, with our method, we were able to perform two classification tasks, achieving an accuracy of 94% (0.94 AUC) for tumor tissue classification and 86% accuracy (0.83 AUC) for partial vs complete response to treatment, outperforming both static radiomics and standard PET image use. We believe that applying radiomics to the time domain has the potential to revolutionize the way we evaluate treatment responses in breast cancer patients. By extracting meaningful features from both static and dynamic PET images, we can more accurately predict patient out-comes and tailor therapeutic interventions to individual needs. This information has been included in the discussion section.

Reviewer 2 Report
Overall, the paper is well put together, with a thorough investigation protocol and remarkable results.
The introduction could potentially benefit from the following suggestions:
- Please consider mentioning more exact data in terms of sensitivity, specificity and accuracy of PET-CT and various imaging agents regarding breast cancer detection and staging.
- I suggest that the authors move the paragraph that references Moland et al’ systematic review in the Discussion section.
The Materials and methods section could be improved by:
- Specifying more clearly the Inclusion and Exclusion criteria on the selected cases.
- Detailing the technical specifications of the employed PET scanners, if available in the used archive.
The Results section could be improved by adding a table that elaborated upon the demographic data of the selected cases.
Lastly, the Discussions section should include how extracting textural features from scans performed on different machines affect the overall accuracy and why haven’t the authors taken into consideration SMOTE or other data augmentation strategies, if faced with such a reduced cohort.
Author Response
Please see the attachment.
Before we begin the point-by-point reply to the reviewer's comments, we would sincerely like to thank the Editor and Editorial Board for the opportunity to resubmit a revised version of our paper, which has been revised in accordance with all reviewer's suggestions.
We would also like to thank the Reviewers for their very insightful comments and feedback on our manuscript. We greatly appreciate your valuable suggestions for enhancing the quality of our research. We have addressed each point as detailed below. We are confident that the revised version of the Manuscript is considerably improved with respect to the first submission and hope that the Reviewers will be available to re-read our work in this respect. All changes in the revised version of the manuscript are highlighted in tracked changes.
Reviewer 2
Overall, the paper is well put together, with a thorough investigation protocol and remarkable results.
Reply: We would like to thank Reviewer 2 for appreciating our work.
The introduction could potentially benefit from the following suggestions:
- Please consider mentioning more exact data in terms of sensitivity, specificity and accuracy of PET-CT and various imaging agents regarding breast cancer detection and staging.
Reply: We thank the reviewer for this comment. In the Introduction, we have mentioned the use of FDG PET tracer, together with FLT. In the revised version of the Manuscript, we have added more detailed information about sensitivity and specificity and we now also mention the clinical validity of additional tracers, such as 18F-FES and 89Zr-trastuzumab.
- I suggest that the authors move paragraph that references Moland et al’ systematic review in the Discussion section.
Reply: The reference to the review by Morland et al has been moved to the Discussion section.
The Materials and methods section could be improved by:
- Specifying more clearly the Inclusion and Exclusion criteria on the selected cases.
Reply: Thank you. This is now specified in paragraph 2.1. In particular, we listed the exclusion/inclusion criteria of the ACRIN 6688 protocol, as also requested by the first reviewer, and specified that our subgroup of patients was selected based on the availability of a dynamic PET scan at baseline.
- Detailing the technical specifications of the employed PET scanners, if available in the used archive.
Reply: From the information found in the archive, in the revised version of the paper we completed paragraph 2.1 by including more information about the PET acquisition. Dynamic PET images were acquired following a bolus injection of 167 MBq (mean; range, 110–204 MBq) using a General Electric (GE)/Philips Medical System PET/CT system. Dynamic scans (matrix dimension: 128 x 128 x 35; voxel dimensions in the x, y and z axis: 3.9, 3.9, 4.2) consisted of 45 timeframes (16 x 5, 7 x 10, 5 x 30, 5 x 60, 5 x 180, 6 x 300 seconds) and a 60-minute acquisition duration (mean, 70 min; range, 50–101 min). PET images were reconstructed using the CT data for attenuation correction with an ordered-subset expectation maximization iterative reconstruction algorithm (2 iterations and 28 subsets). All patients were scanned on calibrated and ACRIN-accredited PET/CT scanners, which underwent image quality review and SUV testing using a uniform phantom.
The Results section could be improved by adding a table that elaborated upon the demographic data of the selected cases.
Reply: We thank the reviewer for this suggestion. The now table has been included.
Lastly, the Discussions section should include how extracting textural features from scans performed on different machines affect the overall accuracy and why haven’t the authors taken into consideration SMOTE or other data augmentation strategies, if faced with such a reduced cohort.
Reply: We thank the reviewer for this comment. We have updated the description of the limitations of our study in the Discussion section stating that we are aware that our work is limited by the extraction of radiomic features from images acquired using different scanners and that data augmentation could potentially improve our results. It should be noted that, given the heterogeneity of our data, data augmentation techniques would also have to be heterogeneous, potentially introducing subjective confounds in final performance comparison.
